# Resistin and IL-15 as Predictors of Invasive Mechanical Ventilation in COVID-19 Pneumonia Irrespective of the Presence of Obesity and Metabolic Syndrome

**DOI:** 10.3390/jpm12030391

**Published:** 2022-03-03

**Authors:** Carles Perpiñan, Laia Bertran, Ximena Terra, Carmen Aguilar, Jessica Binetti, Miguel Lopez-Dupla, Anna Rull, Laia Reverté, Elena Yeregui, Frederic Gómez-Bertomeu, Joaquim Peraire, Teresa Auguet

**Affiliations:** 1Centre d’Atenció Primària (CAP) Sant Pere, Institut Català de la Salud (ICS), 43202 Reus, Spain; cperpinan.tgn.ics@gencat.cat; 2GEMMAIR Research Group, Department of Medicine and Surgery, Faculty of Medicine, Universitat Rovira i Virgili (URV), Institut d’Investigació Sanitària Pere Virgili (IISPV), 43007 Tarragona, Spain; laia.bertran@urv.cat (L.B.); caguilar.hj23.ics@gencat.cat (C.A.); jessica.binetti@gmail.com (J.B.); 3MoBioFood Research Group, Department of Biochemistry and Biotechnology, Faculty of Chemistry, URV, IISPV, 43007 Tarragona, Spain; ximena.terra@urv.cat; 4Internal Medicine Unit, Hospital Universitari de Tarragona Joan XXIII (HJ23), ICS, 43007 Tarragona, Spain; jmlopezdupla.hj23.ics@gencat.cat (M.L.-D.); jjperaire.hj23.ics@gencat.cat (J.P.); 5INIM Research Group, Department of Medicine and Surgery, Faculty of Medicine, URV, IISPV, 43007 Tarragona, Spain; anna.rull@iispv.cat (A.R.); laia.reverte@iispv.cat (L.R.); eyeregui.hj23.ics@gencat.cat (E.Y.); 6Centro de Investigación Biomédica en Red de Enfermedades Infecciosas (CIBERINFEC), 28029 Madrid, Spain; 7Microbiology Unit, Laboratori Clínic Camp de Tarragona, HJ23, ICS, 43007 Tarragona, Spain; ffgomez.hj23.ics@gencat.cat

**Keywords:** COVID-19, SARS-CoV-2, obesity, metabolic syndrome, cytokines, prediction

## Abstract

The cytokine signature present in COVID-19 could provide information on the pathogenic mechanisms of the disease and could identify possible prognostic biomarkers and possible therapeutic targets. In this longitudinal work, we studied the clinical and biochemical parameters and circulating cytokine levels of 146 patients at the time of admission for COVID-19 and 4–6 weeks later. The main objective of this study was to determine whether basal cytokines could be early prognostic biomarkers of COVID-19, and also to analyze the impact of comorbidities, such as obesity or metabolic syndrome (MS), in the cytokine profile. The levels of most inflammatory cytokines were elevated on admission in relation to the level that was reached 4–6 weeks later, except for IL-1β, which was lower on admission; these levels were irrespective of the presence of obesity or MS since the cytokine storm masks these inflammatory processes. Among the cytokines analyzed, those that correlated with a worse prognosis of COVID-19 were resistin, IL-6, IL-8, IL-15, MCP-1 and TNF-α. Specifically, resistin and IL-15 are the best early predictors of requiring invasive ventilation. Therefore, resistin and IL-15 should be included in the personalized treatment decision algorithm of patients with COVID-19.

## 1. Introduction

People infected with severe acute respiratory syndrome coronavirus 2 (SARS-CoV-2) present mild to severe symptoms depending on viral load and individual characteristics [1,2]. Factors associated with the severity and worse prognosis of the ongoing coronavirus disease 2019 (COVID-19) pandemic are age, a compromised immune system and the presence of chronic diseases, such as type 2 diabetes mellitus (T2DM) and arterial hypertension [3]. Indeed, a growing number of clinical reports indicate that obesity is a risk factor for COVID-19 severity [3,4,5,6,7,8,9], especially if the obesity is morbid or severe. In addition, obesity increases the risk of admission to intensive care units (ICUs) and of mortality [10]. The potential molecular mechanisms whereby obesity contributes to the pathogenesis of COVID-19, in addition to the obesity-related deregulated immune response, are chronic inflammation, endothelial imbalance, metabolic dysfunction and its associated comorbidities, and dysfunctional mesenchymal stem cells/adipose-derived mesenchymal stem cells [11]. In addition to obesity, the components of metabolic syndrome (MS) also seem to be associated with severe COVID-19 [12,13,14,15]. It has been suggested that enhanced angiotensin converting enzyme 2 (ACE2) expression, pre-existing endothelial dysfunction and a procoagulant state induced by adipocytokine deregulation in MS may play a crucial role in the development of severe COVID-19 [16].

Regarding adipocytokine deregulation, accumulating evidence suggests that in addition to direct viral damage, uncontrolled inflammation contributes to disease severity in COVID-19 [17]. This is characterized by a deep cytokine response in the host with elevated levels of inflammatory mediators, which culminates in a “cytokine storm” that leads to tissue damage and multiorgan failure [18,19,20]. Several cytokines have been strongly involved in this “cytokine storm”. However, IL-15, a key cytokine involved in lymphocyte activation and homeostasis [21], and resistin, an adipocytokine linked to obesity, insulin resistance, and inflammation [22], have been poorly studied in this regard. Additionally, a combined cytokine panel may improve the accuracy of the predictive value for adverse outcomes beyond standard clinical data alone [23]. Therefore, it would be important to identify specific cytokines that can predict the prognosis of patients affected by COVID-19, especially those suffering from obesity and/or MS. In addition, knowing the fluctuation of these cytokine levels at the beginning of the infection and at the time of convalescence can increase our understanding of the pathogenesis, and it could facilitate the transition towards a precision medicine approach to COVID-19. Highlighting the phenotypes of subgroups that may benefit from targeted immunomodulatory treatments earlier could improve outcomes for these patients and could even be important for developing more effective treatments.

In this sense, the present study aims to characterize the inflammatory cytokine profile at the time of admission and after 4–6 weeks in a cohort of patients with the SARS-CoV-2 infection, confirmed by a polymerase chain reaction (PCR) test and classified according to the World Health Organization (WHO) classification of COVID-19 pneumonia severity to find prognostic biomarkers of COVID-19 severity [24]. Moreover, we aimed to study the impact of comorbidities, such as obesity and MS, on the cytokine profile.

## 2. Materials and Methods

### 2.1. Particiants

The study was approved by the institutional review board (Institut d’Investigació Sanitària Pere Virgili (IISPV) CEIm; 079/2020), and all participants gave informed consent. All patients enrolled in this study (*n* = 146) were diagnosed with SARS-CoV-2 infection and admitted to Internal Medicine Department in University Hospital of Tarragona Joan XXIII between 2 February and 26 September 2020. Throat swab samples were obtained from all patients at admission and tested for SARS-CoV-2 using reverse transcriptase-PCR and real-time quantitative PCR assays.

Patients were admitted by pneumonia (*n* = 126) or by other non-respiratory symptoms with positive SARS-CoV-2 infection (*n* = 20). The severity of the pneumonia will be classified according to the WHO 8-point classification of severe pneumonia by COVID-19 [24] namely: (0) No clinical or virological evidence of COVID-19 infection; (1) Infected without limitations; (2) Limitation of activity; (3) Hospitalized without oxygen therapy; (4) Oxygen by mask or nasal; (5) Non-invasive ventilation (NIV) (continuous positive airway pressure [CPAP] or positive bipressure in the airways [BiPAP]) or high-flow oxygen (HFO)/high-flow nasal cannulas (HFNC); (6) Intubation with mechanical ventilation (MV), mask with reservoir or oxygen with high flow nasal goggles; (7) MV or extracorporeal membrane oxygenation (ECMO), support with vasopressors, dialysis/renal replacement therapy; (8) Death. In order to facilitate understanding, we categorized patients with pneumonia in mild (WHO-3 (presenting pneumonia)), moderate (WHO-4 and 5) and severe (WHO-6, 7 and 8) disease.

Due to the role of obesity and metabolic syndrome in the prognosis of COVID-19 and the underlying inflammatory process in these two metabolic diseases, we also conducted two sub-studies:-Obesity sub-study: the study participants were classified according to their BMI: patients without obesity (BMI < 30 kg/m^2^; *n* = 109) or with obesity (BMI ≥ 30 kg/m^2^; *n* = 37).-Metabolic syndrome sub-study: 124 patients of the whole cohort, which had enough clinical information available, were classified depending on the presence of MS (MS, *n* = 15; non-MS, *n* = 109) according to Alberti et al. criteria [25].

Patients were excluded at initial assessment if they were chronically immunosuppressed, receiving long-term oral corticosteroids, antivirals, hydroxychloroquine, anti-interleukin (IL)-1, anti-IL-6 or anti-tumor necrosis factor (TNF) therapy, known to be pregnant, on dialysis for chronic kidney disease, had active neoplasia, or had a history of vasculitis or connective tissue disease.

### 2.2. Data Collection

Clinical data were recorded prospectively daily. A trained team of physicians collected epidemiological data, past medical history, clinical data, treatments, the treatment established for COVID-19, comorbidities, chest computed tomography (CT) or X-ray results, anthropometrical evaluation (measure of weight, height and BMI calculation) and outcomes for all subjects. Patient confidentiality was protected by assigning an anonymous identification code, and the electronic data were stored in a password-protected computer. 

The following patient characteristics were analyzed: sex, age, BMI, background, comorbidities, radiological findings, clinical characteristics and biochemical parameters. 

Additionally, the whole cohort was enrolled into a longitudinal study, and 4–6 weeks after admission, they were reassessed from the clinical, analytical and radiological point of view if considered, so that each patient was control of himself.

### 2.3. Biochemical Analyses

At the time of admission, all patients were examined in the laboratory, including routine blood cell count and biochemistry, coagulation parameters, c reactive protein and procalcitonin. Blood samples were obtained from all participants. Biochemical parameters were analyzed using a conventional automated analyzer after 12 h of fasting. All serum samples to carry out the present study were collected immediately after hospital admission and 4–6 weeks after admission.

### 2.4. Plasma Measurements

We determined the circulating levels of different molecules related to inflammation, including interleukines such as IL-1β, IL-6, IL-7, IL-8, IL-10, IL-13, IL-15, IL-17A and IL-18, and other adipo/cytokines such as TNF-α, monocyte chemoattractant protein 1 (MCP-1), resistin, interferon (INF)-β and IFN-γ. Circulating levels of IL-1β, IL-6, IL-7, IL-8, IL-10, IL-13, IL-15, IL-17A, IL-18, TNF-α and IFN-γ were determined using multiplex sandwich immunoassays and the MILLIPLEX MAP Human High Sensitivity T Cell Magnetic Bead Panel (HSTCMAG-28SK-07, Millipore, Billerica, MA, USA). Circulating levels of resistin and MCP-1 were measured using MILLIPLEX MAP Human Adipokine Magnetic Bead Panel 1 (HADK1MAG-61K-04, Millipore, Billerica, MA, USA) and MILLIPLEX MAP Human Adipokine Magnetic Bead Panel 2 (HADK2MAG-61K-02, Millipore, Billerica, MA, USA). Circulating levels of IFN-β were measured using Human Cytokine/Chemokine Magnetic Bead Panel IV (HCYP4MAG-64K, Millipore, Billerica, MA, USA). The whole assay was performed using the Bio-Plex 200 instrument, according to the manufacturer’s instructions.

### 2.5. Statistical Analysis

The data were analyzed using the SPSS/PC+ for Windows statistical package (version 23.0; SPSS, Chicago, IL, USA). All results were expressed as mean (standard deviation, SD) or median (interquartile range, IQR) for continuous variables and as counts (percentage) for categorical variables. Continuous variables were compared between groups with the Student t-test or Mann–Whitney U-test, according to their distribution. Paired samples were compared using Kruskall–Wallis or Wilcoxon tests. Categorical variables were compared with the Chi squared or Fisher’s exact test. The strength of the association between variables was calculated using the Spearman ρ correlation test (nonparametric variables). 

To calculate whether certain variables could predict the evolution of the disease (according to the WHO classification), we used the Classification and Regression Tree (CRT) method. Variables included to generate the regression tree were age, gender, BMI, T2DM presence, resistin, IL-8, IL-15 and MCP-1. CRT analysis splits the data into segments that are as homogeneous as possible with respect to the dependent variable. A multivariate analysis was performed to obtain significant and independent risk factors for disease evolution according to the WHO classification. Once the possible confounding factors had been adjusted, a multiple logistic regression analysis was performed, and the adjusted odds ratio (OR) was found with maximum likelihood ratio method and its 95 confident interval (CI)%. For the elaboration of the multivariate model, plausibility due to the current scientific evidence and the level of statistical significance in the bivariate analysis were taken into account. Finally, prediction accuracy was determined using area under the curve (AUC, using the pROC package in R) and Youden index. A *p*-value lower than 0.05 was considered statistically significant. Graphics were elaborated using GraphPad Prism v7.04.

## 3. Results

### 3.1. Baseline Characteristics of Whole Cohort

The main characteristics of the whole studied cohort are described below. The median age was 58 years (interquartile range; IQR 46.75–71.00), BMI 24.31 kg/m^2^ (IQR 26.71–30.30): (BMI < 30 kg/m^2^: 74.5%, BMI ≥ 30 kg/m^2^: 25.5%) and 53.4% were male. The background of the cohort was: 46.4% of patients exercised regularly, 73.9% were non-smokers, 8.5% were active smokers and 17.6% former smokers; 17.1% of patients consumed alcohol significantly. 

The 14.4% of subjects had T2DM, 29.5% presented dyslipidemia, 39.7% had arterial hypertension, 10.3% showed previous cardiovascular diseases, 8.9% previous respiratory disease and 4.1% had a previous history of cancer. Regarding clinical characteristics, 1.4% of patients were asymptomatic from the respiratory point of view (they had been admitted for other unrelated illnesses and were diagnosed through a pre-hospitalization screening program), 21.2% had mild, 56.8% had moderate and 20.5% had critical symptoms. The 86.30% of the patients presented pneumonia, the 54.1% of patients had respiratory failure and 20.5% were admitted to the ICU. The global mortality rate raised up to 4.1%. 

Regarding radiologic characteristics, a bilateral interstitial pattern was observed in chest X-ray or CT in 69.2% of cases, and pleural effusion in only 0.7%. A concomitant pulmonary embolism was present in 1.4% of patients.

The treatment established for COVID-19 was hydroxychloroquine in 38.4% of cases, azithromycin in 43.8%, lopinavir-ritonavir in 34.2%, tozilizumab in 2.7%, INF in 5.5%, corticosteroids in 43.8% and remdesivir in 18.5%. This treatment was changed throughout the period studied, according to the clinical guidelines, which were frequently updated. 

In regard to oxygen therapy, 54.1% of patients needed oxygen by mask or nasal, 13% HFO or HFNC, 0.7% non-invasive ventilation and continuous positive airway pressure (NIV–CPAP), and 0.7% non-invasive ventilation and bilevel positive airway pressure (NIV–BiPAP); 17.8% of patients needed MV by intubation, and 4.8% by mask with reservoir; 12.3% of patients needed vasopressors with MV and 3.4% dialysis.

### 3.2. Baseline Cytokine Storm Related to COVID-19 Pneumonia Severity

First, we evaluated the clinical characteristics of the whole cohort according to the presence of mild, moderate or severe pneumonia (Table 1). We have classified in mild pneumonia when WHO was 3, moderate in WHO 4–5 and severe in WHO 6–7–8. The groups were comparable in terms of age and gender. To note, we have not found differences in pneumonia severity depending on the presence of obesity or metabolic syndrome. 

Then, we evaluated the baseline circulating levels of cytokines of these patients, and we observed that resistin, IL-6, IL-8, IL-15, MCP-1 and TNF-α levels were significant different between groups. These differences were graphically represented in Figure 1, where we can see that circulating levels of these cytokines were increased between mild and moderate–severe pneumonia.

### 3.3. Biochemical Parameters and Cytokine Levels of the Whole Cohort, at Baseline and 4–6 Weeks after Admission and Clinical Residual Manifestations at 4–6 Weeks

Then, in the longitudinal study, we compared clinical characteristics, biochemical parameters and cytokine levels at the moment of admission, and at 4–6 weeks later, in order to assess the level of cytokines during the recovery from the disease. At 4–6 weeks of enrolment, 65.1% of patients were asymptomatic and the rest of the 34.9% of patients presented as main clinical symptoms: respiratory symptomatology (13.0%), neurological symptomatology (6.8%) and asthenia or fatigue (15.1%). 

In Table 2, biochemical parameters and cytokine levels of the whole cohort at the baseline and at 4–6 weeks after admission are shown. Of note, the lymphocyte count, total-cholesterol and alkaline phosphatase (AP) were lower at the baseline, whereas ferritin, C-reactive protein (CRP), aspartate aminotransferase (AST), gamma-glutamyltransferase (GGT) and lactate dehydrogenase (LDH) were significantly higher at the baseline compared to 4–6 weeks after admission. 

Regarding cytokines, resistin, INF-γ, IL-6, IL-7, IL-8, IL-13, IL-15, IL-17A, IL-18 and TNF-α decreased significantly after 4–6 weeks compared to baseline levels at the acute phase of the disease. Meanwhile, IL-1β levels increased significantly after 4–6 weeks (Table 2).

### 3.4. Correlations between Baseline Biochemical Parameters and Cytokine Levels and the Clinical Residual Manifestations at 4–6 Weeks of the COVID-19 Cohort

To analyze the possible predictive potential of biochemical parameters and the cytokine profile, we assessed Spearman correlations between baseline levels of some biochemical parameters and cytokine levels with clinical manifestations at 4–6 weeks after the admission. We found that D-dimer levels negatively correlate and AST levels positively correlate with the persistence of COVID-19 respiratory symptoms one month later. Moreover, we found that lymphocytes and ALT levels correlate positively with the presence of asthenia or fatigue at 4–6 weeks; meanwhile CRP, troponin, resistin, IL-8 and MCP-1 levels correlate negatively (data not shown).

### 3.5. Biochemical Parameters and Cytokine Levels of the COVID-19 Cohort Classified according to the BMI (<30 and ≥30 kg/m^2^) and according to the Presence of MS

Table 3 shows differences in baseline characteristics, biochemical variables and cytokine levels between patients with or without obesity. Patients with obesity (BMI ≥ 30 kg/m^2^, 25.5% of the general cohort) or without obesity (BMI < 30 kg/m^2^, 74.5%) were comparable in terms of age and gender. At the baseline, lymphocyte levels were lower in patients without obesity compared to patients with obesity, but we could not find significant differences regarding other biochemical values, and neither for cytokine levels. When we compared the risk of needing invasive ventilatory support, ICU admission or mortality between groups, we did not find significant differences.

Then, we studied the differences in the baseline characteristics, biochemical variables and cytokine levels between patients with or without MS (Table 4). Patients with MS (12.1% of the general cohort) were older, with a higher BMI than non-MS patients. With regard to biochemical parameters, MS patients showed higher levels of fasting glucose, compared to non-MS patients. The only cytokine that showed significant differences between both subgroups was the baseline IL-1β levels, that were lower in MS than non-MS patients. When we compared the risk of needing invasive ventilatory support, ICU admission or mortality between non-MS subjects and MS patients, we did not find significant differences.

### 3.6. Correlations between the Baseline Levels of Cytokines with Clinical Characteristics of Patients with COVID-19

Then, we wanted to analyze the relationship between the baseline cytokine concentrations with general characteristics, and clinical and biochemical variables at the time of hospital admission. These results are described in the Appendix A. The most relevant results are that resistin, IL-6, IL-8, IL-15, IL-18, MCP-1 and TNF-α are the main cytokines associated to COVID-19 symptomatology; meanwhile resistin, IL-8, IL-15, MCP-1 and TNF-α used to be associated with oxygen therapy necessity.

### 3.7. Predictive Value of Cytokine Levels for COVID-19 Outcome

COVID-19 can be conspicuously phasic, with major deteriorations in some patients occurring between 7–12 days post-symptom onset. In this sense, there is intense interest in early patient stratification [26]. Therefore, the last objective of the present work was to study whether baseline cytokine levels were capable of predicting the evolution of disease according to the WHO classification of severe pneumonia by COVID-19. First, we used the CRT method to obtain the best cut-off values of each cytokine to predict COVID-19 pneumonia severity. We created a decision tree with the general cohort of the study, entering, on the one hand, clinical variables regarding pneumonia categorization (mild, moderate and severe) and, on the other hand, the baseline cytokine levels as independent variables. The main results are shown in Figure 2. 

Figure 2 showed that patients that presented IL-8 levels lower than 20.55 pg/mL would not develop severe pneumonia (WHO 6, 7 and 8). When patients have higher levels of IL-8, we need to observe resistin: on the one hand, the 73.1% of patients with resistin levels higher than 70.378 × 10^3^ pg/mL would present severe pneumonia; on the other hand, to evaluate patients with resistin levels lower than the cut-off, we need to observe IL-15. In this sense, the 75% of patients with IL-15 levels lower than 5.845 pg/mL would need non-invasive ventilatory support, and the 25% would not need oxygen therapy, which means that the 100% of patients would not need invasive ventilatory support. 

Additionally, we wanted to perform another decision tree to evaluate the prediction of the studied cytokines according to concrete WHO classification (WHO-3, 4, 5, 6, 7 and 8). Results were represented graphically in Appendix A. Appendix A showed that it was expected that 96.6% of patients with a resistin level lower than 33.294 × 10^3^ pg/mL would not need NIV (WHO-5). Figure 2 reported, on the one hand, that 91.8% of patients with resistin levels lower than 54.2278 × 10^3^ pg/mL would not need MV by intubation (WHO-6) and, also, 100% of these patients with an IL-8 level lower than 25.1 pg/mL would not need it. In Figure 2, the classification tree analysis revealed that 98.6% of patients with resistin levels lower than 51.9693 × 10^3^ pg/mL and MCP1 lower than 830.87 pg/mL would not need MV, nor support by vasopressors or dialysis (WHO-7). Finally, in Figure 2, we observed that 100% of patients with resistin levels lower than 54.22784 × 10^3^ pg/mL would not die by COVID-19. We did not find a relevant prediction for WHO-4.

Then, to perform a logistic regression analysis for risk evaluation, we categorized the continuous variables resistin, IL-8 and IL-15 into high or low levels of each cytokine according to the cut-off values obtained in the regression tree. For this analysis, we also classified patients according to COVID-19 severity in a binary variable: non-invasive ventilation (WHO-4 and WHO-5) vs. invasive ventilation (WHO-6 and WHO-7). The analysis revealed that patients with high levels of resistin and IL-15 were at a high risk of suffering more severe COVID-19 symptoms; in concrete, they would more probably require invasive ventilation. In this sense, high resistin levels increased 4.4 times the risk of requiring MV compared with those patients who presented low resistin levels (*p* = 0.005, IC 95% = 1.565–12.409). Regarding IL-15, high levels of this IL increased 2.7 times such risk (*p* = 0.048, IC 95% = 1.008–7.710).

Additionally, we classified patients according to another binary variable: ICU admission or not. The analysis showed that patients with high levels of resistin and IL-8 were at high risk of ICU admission. In this sense, high resistin levels increased 3.8 times the risk of requiring ICU admission compared with those patients who presented low levels (*p* = 0.004, IC 95% = 1.5–9.5). With regard to IL-8, high levels of this cytokine increased 17.2 times such risk (*p* = 0.006, IC 95% = 2.2–133.4).

To further evaluate the potential of these cytokines to be useful in a decision model to assess the COVID severity (according to the need for oxygen therapy or not), we performed ROC curves, as shown in Figure 3. The results indicated that resistin could predict the clinical outcome of COVID-19 patients regarding the need of MV (Figure 3A, *p* < 0.001). Moreover, IL-8 and IL-15 presented high AUC values (Figure 3B, *p* = 0.002 and Figure 3C, *p* = 0.009; respectively). Then, we also aimed to analyze the potential of a model including all these cytokines. For this purpose, we created a combined predictive variable using a logistic regression. It included resistin, IL-15 and IL-8 as independent variables and non-invasive ventilation (WHO-4 and WHO-5) vs. invasive ventilation (WHO-6 and WHO-7) as a binary dependent variable (Figure 3D, *p* < 0.001). The results indicated that the performance of these cytokines as a combined model was lower than resistin alone. Finally, we also included in the combined predictive variable the age, gender and MS presence (AUC: 0.718, *p* < 0.001), however, resistin alone was still the better predictor.

In order to give more relevant clinical results, we have calculated the predetermined Youden index, which represents the best ratio sensitivity/specificity (shown in Figure 3). However, as a false negative is worse and more costly than a false positive in this disease, a responsible clinical decision about the trade-offs is necessary. Therefore, a lower cut-off has to be chosen to increase the sensitivity. Regarding resistin, the best cut-off was 33.403 × 10^3^ pg/mL (37.7% Youden index) with a sensitivity of 88.9% and a specificity of 48.8%. The best cut-off of IL-15 was 6.62 pg/mL (35.5% Youden index) with a sensitivity of 94.3% and a specificity of 41.5%.

## 4. Discussion

The novelty of this work lies in the fact that it is a longitudinal study, by which we could analyze the clinical and biochemical parameters and cytokine levels of patients at the time of admission for COVID-19 and 4–6 weeks later, and we could also use the levels of baseline cytokines to predict COVID-19 pneumonia evolution. The main results are that the levels of inflammatory cytokines decreased considerably at 4–6 weeks compared to at the time of admission in the acute phase of COVID-19, except for IL-1β, which increased at 4–6 weeks after admission. Of note, these levels were not modified by the presence of obesity or MS. The cytokine levels that correlated with a worse prognosis of COVID-19 pneumonia were resistin, IL-6, IL-8, IL-15, MCP-1 and TNF-α. Finally, regarding the prediction of COVID-19 evolution, IL-15 was the best predictor of the necessity of MV and of MV or ECMO plus support with vasopressors and dialysis/renal replacement therapy. Additionally, patients with high levels of resistin and IL-15 are at high risk of requiring invasive ventilation. Moreover, ROC curves revealed that resistin was the best predictor of MV necessity in COVID-19 patients.

In our longitudinal study, as expected, the cytokine levels decreased significantly after 4–6 weeks compared to the baseline levels in the acute phase of the disease, except for IL-1β, which increased at 4–6 weeks after admission. In this sense, the presence of a “cytokine storm” in the COVID-19 acute phase has been described thoroughly in the literature. Its pathology is plausibly linked to the hyperinflammatory response of the body characterized by pathological cytokine levels, and its presence is associated with a greater severity of the disease [27]. The concept of an uncontrolled, cytokine-mediated response was first described in relation to malaria and sepsis [28,29] and subsequently in other diseases, either infectious and noninfectious [30,31,32,33,34,35], or even triggered by certain drugs [36,37,38]. In addition, cytokines and chemokines have long been thought to play an important role in immunity and immunopathology during virus infections. A rapid and well-coordinated innate immune response is the first line of defense against viral infections, but dysregulated and excessive immune responses may cause immunopathology [39,40]. Specifically, in COVID-19, the viral infection is able to trigger an excessive immune response in predisposed individuals, that presents a hyperinflammation state, able to determine tissue and vascular damage. An explosive production of proinflammatory cytokines such as TNF-α, IL-1β and others occurs, greatly exaggerating the generation of molecule-damaging reactive oxygen species [27]. Infected cells activate large numbers of leucocytes, including B and T cells, natural killer cells, macrophages, dendritic cells and monocytes, releasing inflammatory cytokines, which activate more white blood cells, perpetuating the cycle and spreading systemically [41]. With regard to IL-1β, a proinflammatory cytokine that is activated and secreted upon the activation of the inflammasome [42], it has been observed in the peripheral blood of patients with COVID-19-induced pneumonia [17,27]. However, we obtained an unexpected result: IL-1β, tended to be higher at the follow-up. These results probably can be explained because IL-1β has a short half-life in serum and is rarely isolated in peripheral blood [43]. So, our results could not be reliable enough for this reason. 

We also described the correlation of inflammatory cytokine levels (resistin, IL-6, IL-8, IL-15, MCP-1 and TNF-α) with clinical variables related to a worse COVID-19 prognosis. In this sense, other authors reported that critically ill patients admitted to the ICU had higher systemic levels of IL-2, IL-7, IL-10, granulocyte-colony-stimulating factor, INF-γ-inducible protein 10 (IP-10), MCP-1, macrophage inflammatory protein-1A (MIP-1A) and TNF-α [44]. Moreover, it has been suggested that there is a significant association between severe uncontrolled inflammation and mortality [45].

Regarding the prediction of the COVID-19 outcome, using different methods, we found that IL-15 and resistin were the best predictors of the requirement of invasive ventilation. In this sense, it has been extensively described that IL-6 levels, along with TNF-α and IL-10, were a reliable indicator of disease severity and predictive in terms of ventilation support [30,46]. Additionally, Lucas et al. showed that IL-7, IL-15 and IL-2 were increased in COVID-19 and correlated with disease severity [47] and may promote IFN-γ production in an antigen-independent manner [48]. Specifically, in two recent studies, it was found that circulating levels of IL-15 have been implicated as a contributory factor to hospitalization time, disease severity, and mortality in some settings [21,49].

Finally, Dorgham K et al. recently described that distinct cytokine profiles were differentially predictive of mortality according to oxygen support modalities. Thus, increased levels of IFN-α and IFN-β predicted no need for MV; increased TNF-α, IL-10, and INF-α levels predicted the need for MV; and increased IL- 10 and decreased IL-17 and IL-18 levels predicted the need for ECMO [50]. With regard to resistin, Meizlish et al. described elevated levels of this molecule in COVID-19 patients with a fatal evolution. They stated that the detection of this protein, in addition to lipocalin 2 and matrix metallopeptidase 8, at high concentrations, in the circulation of critically ill COVID-19 patients strongly suggests neutrophil activation and degranulation. They hypothesized that components of the neutrophil activation signature described had effector functions that might be detrimental to patients with COVID-19 [22]. In addition, Sundén-Cullberg et al. showed that resistin was a marker of the severity of disease and possibly a mediator of the prolonged inflammatory state in septic shock/severe sepsis [51].

Moreover, in our work, since obesity and MS are associated with a certain degree of chronic underlying inflammation, we wanted to study the relationship between the baseline cytokine levels and the presence of obesity and/or MS in this cohort of COVID-19 patients. However, we did not find any relationship in this regard. Therefore, based on our results, we hypothesize that the poor prognosis presented by obese or MS patients when suffering from COVID-19 pneumonia would be further explained by factors other than the “cytokine storm” that is common to all patients, such as a defective immune response, chronic inflammation, dysfunctional adipose tissue, compromised ciliated airway epithelial cells, mechanical defects, and pulmonary arterial hypertension that impair the lung defense system against the SARS-CoV-2 infection [9].

It must be considered that this work was conducted on a relatively small number of patients in specific groups, such as MS. Additionally, as patients of our study were treated in accordance with current guidelines, they could receive at most one dose of dexamethasone and/or remdesivir prior to admission analysis, therefore, small variations in basal cytokine levels cannot be ruled out. Hence, further studies would be useful to validate these findings.

## 5. Conclusions

Resistin and IL-15 are the best early predictors of invasive ventilation in COVID-19 pneumonia. Therefore, this cytokine profile should be included in the personalized treatment decision algorithm of patients with COVID-19. On the other hand, COVID-19 produces such a severe cytokine storm at the acute phase of the disease that it overshadows the underlying chronic inflammation present in obesity and metabolic syndrome.

## Figures and Tables

**Figure 1 jpm-12-00391-f001:**
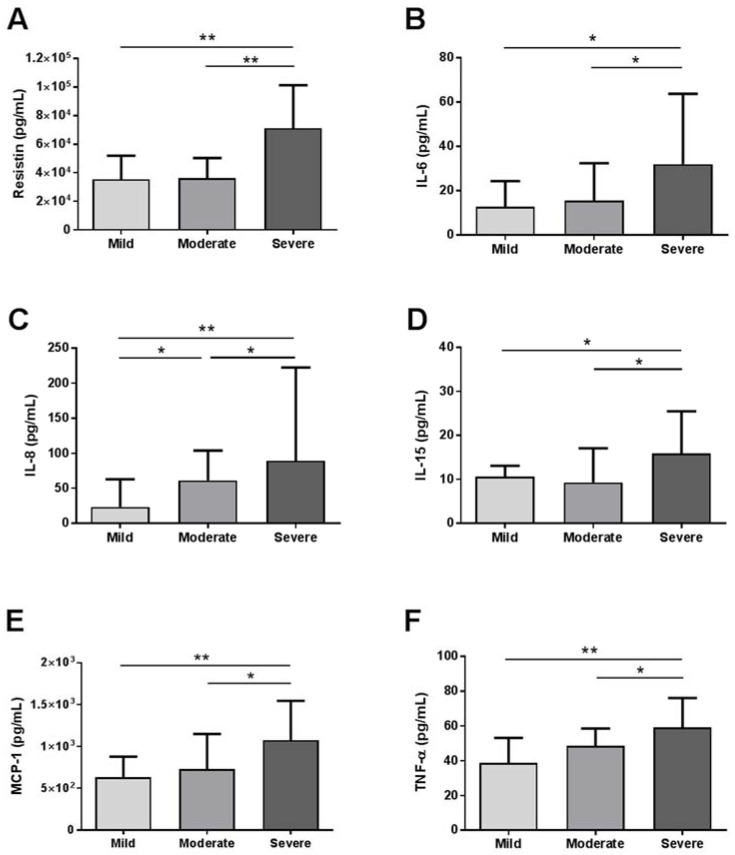
Baseline circulating levels of cytokines of the whole cohort according to COVID-19 pneumonia severity. Serum levels of (**A**) resistin, (**B**) IL-6, (**C**) IL-8, (**D**) IL-15, (**E**) MCP-1 and (**F**) TNF-α in patients with mild, moderate or severe COVID-19 pneumonia. IL, interleukin; TNF-α, tumor necrosis factor alpha; MCP-1, monocyte chemoattractant protein 1. Data were presented as Median and IQR. * *p* < 0.05 and ** *p* < 0.001 were considered statistically significant by Mann–Whitney.

**Figure 2 jpm-12-00391-f002:**
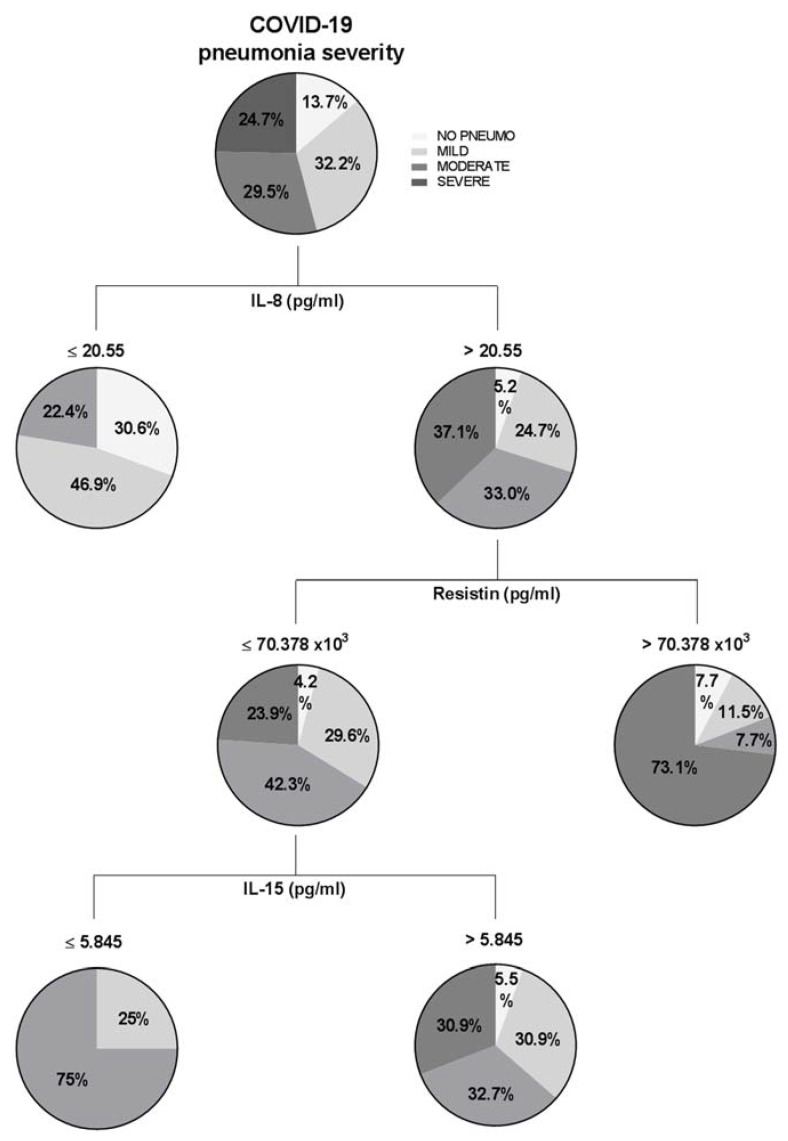
Classification and regression trees of the whole cohort of study were elaborated through CRT method for pneumonia categorization (no pneumonia, mild, moderate and severe). IL, interleukin.

**Figure 3 jpm-12-00391-f003:**
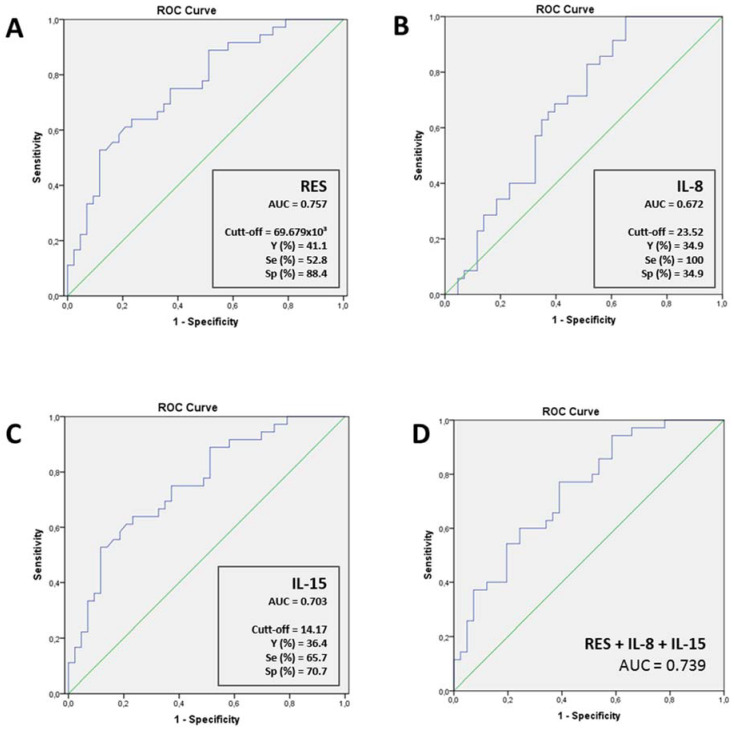
Receiver operating characteristic (ROC) curve and Youden index for (**A**) resistin (*n* = 79, AUC CI95%: 0.64–0.85), (**B**) IL-8 (*n* = 78, AUC CI95%: 0.56–0.77), (**C**) IL-15 (*n* = 78, AUC CI95%: 0.59–0.80) and (**D**) combination of resistin, IL-8 and IL15 (*n* = 70, AUC CI95%: 0.60–0.82) for the prognosis of COVID-19 severity outcomes (need for mechanical ventilation). AUC, area under the curve; IL, interleukin; Y, Youden index; Se, sensitivity; Sp, Specificity.

**Table 1 jpm-12-00391-t001:** Clinical characteristics according to pneumonia severity.

	Pneumonia Severity			
Variables	Mild (WHO 3) *n* = 47	Moderate (WHO 4–5) *n* = 43	Severe (WHO 6–7–8) *n* = 36	*p*-Value
Gender—*n* (%):				
Male	21 (44.70)	22 (51.20)	25 (69.40)	0.074
Age—mean (SD):				
	57.70 (18.16)	62.19 (13.33)	60.64 (11.49)	0.478
Admission stay—mean (SD):				
	7.83 (9.73)	8.79 (5.40)	34.47 (23.83)	<0.001 *
Comorbidities—*n* (%):				
Obesity	10 (23.80)	13 (31.70)	9 (26.50)	0.717
Metabolic syndrome	7 (15.90)	2 (8.30)	4 (11.10)	0637
Diabetes mellitus	7 (14.90)	5 (11.60)	9 (25.00)	0.263
Hypertension	16 (34.00)	26 (60.50)	11 (30.60)	0.011 *
Dyslipemia	15 (31.90)	12 (27.90)	12 (33.30)	0.861
Cardiovascular disease	5 (10.60)	4 (9.30)	3 (8.30)	0.936
Death—*n* (%)				
	0 (0)	0 (0)	6 (16.70)	<0.001 *

SD, standard deviation. Quantitative data are expressed as the mean (SD). Categorical variables are expressed as counts (valid percentage). *p* values were calculated by Kruskal–Wallis test, * Significant differences between different groups of pneumonia severity (*p* < 0.05).

**Table 2 jpm-12-00391-t002:** Biochemical parameters and cytokine levels of the whole cohort at baseline and at 4–6 weeks after hospital admission.

FOLLOW-UP					
Variables	Baseline Mean (SD) or Median (IQR) *n* = 146		4–6 Weeks Mean (SD) or Median (IQR) *n* = 146		*p*-Value
Leukocytes (×10^9^/L)	6.49	(4.89–8.19)	6.14	(4.95–7.49)	0.734
Lymphocytes (×10^9^/L)	1.10	(0.59)	1.92	(0.76)	<0.001 *
Hemoglobin (g/dL)	12.95	(11.62–13.9)	12.70	(11.60–13.62)	0.139
D-Dimer (ng/mL)	540.00	(382.75–888.25)	434.00	(294.00–683.00)	0.070
ESR (mm)	65.15	(37.22)	48.83	(53.89)	0.655
Ferritin (ng/mL)	385.00	(160.00–606.00)	153.00	(58.00–248.25)	<0.001 *
CRP (mg/dL)	7.00	(3.00–14.00)	0.40	(0–0.70)	<0.001 *
Glucose (mg/dL)	103.00	(87.00–128.00)	93.00	(85.00–110.25)	0.054
Total-cholesterol (mg/dL)	140.61	(35.39)	173.51	(50.64)	0.005 *
Creatinine (mg/dL)	0.75	(0.59–0.89)	0.79	(0.62–0.89)	0.160
AST (U/L)	31.00	(23.25–44.00)	23.00	(18.00–30.00)	<0.001 *
ALT (U/L)	28.00	(19.25–51.75)	26.00	(19.00–46.00)	0.332
GGT (U/L)	45.00	(26.00–79.00)	31.00	(20.00–56.00)	0.003 *
AP (U/L)	66.00	(51.75–89.25)	70.00	(59.00–84.00)	0.005 *
LDH (U/L)	285.60	(80.14)	197.35	(42.70)	<0.001 *
Cytokines—Median (IQR):					
Resistin (×10^3^ pg/mL)	38.86	(25.67–67.69)	23.69	(17.73–35.68)	<0.001 *
INF-β (pg/mL)	889.50	(333.14–2230.71)	1080.17	(435.38–2289.21)	0.779
INF-γ (pg/mL)	11.83	(5.50–23.86)	5.95	(3.05–14.99)	<0.001 *
IL-1β (pg/mL)	4.91	(2.22–12.50)	4.94	(2.03–15.49)	<0.001 *
IL-6 (pg/mL)	11.70	(4.90–26.42)	2.90	(2.10–5.30)	<0.001 *
IL-7 (pg/mL)	7.38	(3.93–13.50)	3.61	(1.98–6.95)	<0.001 *
IL-8 (pg/mL)	40.68	(15.88–94.56)	11.14	(7.03–23.40)	<0.001 *
IL-10 (pg/mL)	31.12	(8.97–57.53)	22.23	(9.61–92.33)	0.109
IL-13 (pg/mL)	14.93	(9.45–28.17)	12.86	(7.56–22.19)	<0.001 *
IL-15 (pg/mL)	10.19	(6.28–17.20)	6.42	(4.17–10.34)	<0.001 *
IL-17A (pg/mL)	11.16	(2.90–25.53)	9.73	(3.48–27.97)	0.013 *
IL-18 (pg/mL)	58.06	(33.41–89.35)	27.52	(14.36–43.54)	<0.001 *
MCP-1 (pg/mL)	731.42	(528.77–1047.76)	775.77	(605.63–932.75)	0.826
TNF-α (pg/mL)	46.42	(27.22–61.52)	30.77	(18.33–47.89)	<0.001 *

SD, standard deviation. Quantitative data are expressed as the mean (SD). Categorical variables are expressed as counts (percentage). *p* values were calculated by Kruskal–Wallis test, * Significant differences between different groups of pneumonia severity (*p* < 0.05). ESR, erythrocyte sedimentation rate; CRP, c-reactive protein; AST, aspartate aminotransferase; ALT, alanine aminotransferase; GGT, gamma-glutamyltransferase; AP, alkaline phosphatase; LDH, lactate dehydrogenase; INF, interferon; IL, interleukin; MCP-1, monocyte chemoattractant protein-1; TNF-α, tumor necrosis factor alpha. Quantitative data are expressed as the mean (SD) or median (interquartile range, IQR) depending on the distribution of the variables. * Significant differences between baseline and 4–6 weeks determined by Wilcoxon test (*p* < 0.05).

**Table 3 jpm-12-00391-t003:** Characteristics, baseline biochemical variables and cytokine levels of the COVID-19 cohort for the obesity sub-study.

Variables		WOB Mean (SD); *n* (%) *n* = 109	OB Mean (SD); *n* (%) *n* = 37	*p*-Value
Age (years)		58.43 (15.65)	53.46 (15.17)	0.091
Sex N (%)	Male	57 (55.9)	14 (40.0)	0.106
BMI (kg/m^2^)		25.27 (2.90)	34.99 (4.65)	<0.001 *
SBP (mmHg)		128.75 (19.90)	131.37 (18.04)	0.409
DBP (mmHg)		78.82 (9.96)	80 (71–88.25)	0.104
Admission stay—mean (SD)		15.26 (17.16)	12.70 (16.43)	0.502
ICU admission N (%)		21 (20.60)	7 (20.00)	0.941
Invasive ventilatory support need N (%)		25 (24.5)	9 (25.7)	0.402
Mortality N (%)		5 (4.90)	1 (2.90)	0.611
Biochemical parameters—Mean (SD) or Median (IQR):				
Leukocytes (×10^9^/L)		6.51 (4.70–8.19)	6.09 (5.20–7.68)	0.998
Lymphocytes (×10^9^/L)		1.02 (0.56)	1.31 (0.63)	0.009 *
Hemoglobin (g/dL)		13.00 (11.55–13.80)	12.80 (12.00–14.05)	0.437
D-Dimer (ng/mL)		541 (367.00–1047.00)	554.5 (411.00–746.00)	0.751
ESR (mm)		68.05 (36.56)	54.23 (35.88)	0.083
Ferritin (ng/mL)		396 (154.00–701.00)	351.5 (154.50–457.50)	0.317
CRP (mg/dL)		6.65 (2.82–13.42)	6.95 (3.10–14.72)	0.681
Glucose (mg/dL)		103.5 (88.00–128.75)	103 (84.00–130.00)	0.954
Total-cholesterol (mg/dL)		141.44 (38.50)	140.35 (29.27)	0.718
Creatinine (mg/dL)		0.77 (0.58–0.89)	0.70 (0.61–0.88)	0.685
AST (U/L)		30 (23.50–43.50)	31 (23.50–44.50)	0.583
ALT (U/L)		28 (19.00–53.00)	33 (23.50–51.50)	0.271
GGT (U/L)		44 (24.00–76.00)	54 (33.00–87.00)	0.058
AP (U/L)		66 (49.00–90.00)	67 (53.00–90.75)	0.666
LDH (U/L)		280.25 (86.97)	293.64 (55.43)	0.138
Troponin (ng/L)		6 (2.00–14.25)	4 (3.00–14.00)	0.675
Cytokines—Median (IQR):				
Resistin (×10^3^ pg/mL)		36.62 (21.37–54.32)	35.43 (29.26–69.95)	0.225
INF-β (pg/mL)		1386.98 (568.33–2230.71)	299.07 (228.25–2497.14)	0.184
INF-γ (pg/mL)		11.69 (5.20–23.83)	8.78 (5.87–20.56)	0.846
IL-1β (pg/mL)		5.80 (2.63–13.17)	4.50 (1.88–14.98)	0.645
IL-6 (pg/mL)		8.90 (4.78–23.65)	11.70 (5.22–16.52)	0.957
IL-7 (pg/mL)		8.12 (3.56–14.13)	6.61 (3.91–11.96)	0.828
IL-8 (pg/mL)		37.81 (13.25–86.42)	68.61 (21.73–127.80)	0.076
IL-10 (pg/mL)		50.30 (28.72–90.48)	35.88 (7.66–55.53)	0.979
IL-13 (pg/mL)		14.79 (8.96–27.36)	16.44 (11.75–34.77)	0.462
IL-15 (pg/mL)		10.83 (6.30–19.71)	7.73 (5.49–14.27)	0.137
IL-17A (pg/mL)		11.04 (2.29–30.41)	11.16 (3.48–20.52)	0.797
IL-18 (pg/mL)		50.30 (28.72–90.48)	60.94 (40.4–79.69)	0.302
MCP-1 (pg/mL)		725.81 (505.47–1021.62)	751.19 (514.62–1077.91)	0.642
TNF-α (pg/mL)		43.85 (25.42–58.66)	53.20 (27.49–70.01)	0.132

WOB, patients without obesity; OB, patients with obesity; SD, standard deviation; BMI, body mass index; SBP, systolic blood pressure; DBP, diastolic blood pressure; ESR, erythrocyte sedimentation rate; CRP, c-reactive protein; AST, aspartate aminotransferase; ALT, alanine aminotransferase; GGT, gamma-glutamyltransferase; AP, alkaline phosphatase; LDH, lactate dehydrogenase; IL, interleukin; MCP-1, monocyte chemoattractant protein 1; TNF-α, tumor necrosis factor alpha. Quantitative data are expressed as the mean (SD) or median (interquartile range, IQR), depending on the distribution of the variables. Categorical variables are expressed as counts (percentage). * Significant differences between WOB group and OB group (*p* < 0.05) by Mann–Whitney.

**Table 4 jpm-12-00391-t004:** Characteristics, baseline biochemical variables and cytokine levels of the COVID-19 cohort used for the sub-study of MS classified according to the Alberti et al. criteria [25].

Variables		Non-MS Mean (SD); *n* (%) *n* = 109	MS Mean (SD); *n* (%) *n* = 15	*p*-value
Age (years)		55.60 (16.63)	68.80 (9.26)	0.003 *
Sex N (%)	Male	59.00 (54.10)	9.00 (60.00)	0.670
BMI (kg/m^2^)		27.20 (5.49)	30.93 (4.48)	0.002 *
SBP (mmHg)		128.64 (20.56)	134.20 (15.38)	0.212
DBP (mmHg)		79.45 (10.35)	78.53 (10.22)	0.615
Admission stay—mean (SD)		17.51 (19.43)	13.71 (20.67)	0.432
ICU admission N (%)		28 (25.70)	2 (13.30)	0.297
Invasive ventilatory support need N (%)		32 (29.40)	4 (26.70)	0.579
Mortality N (%)		5 (4.60)	1 (6.70)	0.726
Biochemical parameters—Mean (SD) or Median (IQR):				
Leukocytes (×10^9^/L)		6.55 (4.85–8.56)	6.68 (5.59–7.60)	0.981
Lymphocytes (×10^9^/L)		1.07 (0.60)	1.14 (0.61)	0.631
Hemoglobin (g/dL)		12.95 (11.62–13.90)	12.40 (10.75–14.25)	0.456
D-Dimer (ng/mL)		545.50 (389.50–888.25)	554.50 (313.00–813.75)	0.634
ESR (mm)		66.91 (36.78)	59.33 (48.37)	0.402
Ferritin (ng/mL)		391.50 (147.25–685.75)	396.00 (142.00–492.50)	0.855
CRP (mg/dL)		7.10 (3.00–14.00)	6.45 (2.12–17.75)	0.748
Glucose (mg/dL)		103.00 (86.00–123.00)	130.00 (102.75–150.75)	<0.001 *
Total-cholesterol (mg/dL)		135.23 (33.97)	141.90 (33.83)	0.462
Creatinine (mg/dL)		0.75 (0.58–0.89)	0.86 (0.73–0.98)	0.054
AST (U/L)		31.50 (25.00–44.75)	22.50 (19.50–53.75)	0.272
ALT (U/L)		30.50 (20.25–54.00)	25.50 (16.00–63.75)	0.484
GGT (U/L)		44.00 (27.00–79.00)	55.00 (23.75–129.50)	0.406
AP (U/L)		67.00 (53.00–89.00)	64.00 (48.00–115.50)	0.682
LDH (U/L)		192.90 (42.02)	212.08 (59.68)	0.835
Troponin (ng/L)		6.00 (2–14.25)	10.50 (4.25–27.75)	0.094
Cytokines—Median (IQR):				
Resistin (×10^3^ pg/mL)		41.22 (23.52–71.23)	39.76 (32.58–69.95)	0.878
INF-β (pg/mL)		889.50 (333.15–2466.04)	996.43 (299.07-)	0.565
INF-γ (pg/mL)		11.40 (5.35–21.77)	15.35 (5.37–25.69)	0.825
IL-1β (pg/mL)		5.89 (2.43–13.17)	2.49 (1.09–5.47)	0.035 *
IL-6 (pg/mL)		11.8 (5.10–28.15)	15.60 (3.00–29.10)	0.905
IL-7 (pg/mL)		7.43 (3.36–14.12)	6.03 (3.95–10.07)	0.712
IL-8 (pg/mL)		40.55 (14.41–98.12)	68.61 (27.97–94.56)	0.338
IL-10 (pg/mL)		35.03 (8.19–65.49)	24.42 (4.98–50.54)	0.474
IL-13 (pg/mL)		15.48 (8.96–30.80)	12.03 (5.15–19.05)	0.141
IL-15 (pg/mL)		10.73 (6.88–17.52)	9.90 (6.57–15.88)	0.594
IL-17A (pg/mL)		11.39 (2.44–25.53)	3.48 (2.29–11.40)	0.103
IL-18 (pg/mL)		52.50 (24.62–81.20)	68.81 (47.18–103.74)	0.203
MCP-1 (pg/mL)		753.29 (559.36–1071.87)	829.87 (392.41–954.92)	0.449
TNF-α (pg/mL)		44.96 (26.33–65.54)	44.96 (36.10–57.63)	0.837

Non-MS, non-metabolic syndrome patients; MS, metabolic syndrome patients; SD, standard deviation; BMI, body mass index; SBP, systolic blood pressure; DBP, diastolic blood pressure; ESR, erythrocyte sedimentation rate; CRP, c-reactive protein; AST, aspartate aminotransferase; ALT, alanine aminotransferase; GGT, gamma-glutamyltransferase; AP, alkaline phosphatase; LDH, lactate dehydrogenase; IL, interleukin; MCP-1, monocyte chemoattractant protein 1; TNF-α, tumor necrosis factor alpha. Quantitative data are expressed as the mean (SD) or median (interquartile range, IQR) depending on the distribution of the variables. * Significant differences between non-MS group and MS group (*p* < 0.05) by Mann–Whitney.

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
