# Peer review of "Resistin and IL-15 as Predictors of Invasive Mechanical Ventilation in COVID-19 Pneumonia Irrespective of the Presence of Obesity and Metabolic Syndrome"

_jpm, 2022, doi:10.3390/jpm12030391_

Round 1

Reviewer 1 Report

The manuscript entitled “Resistin and IL-15 as predictors of mechanical ventilation in COVID-19 pneumonia irrespective of the presence of obesity and metabolic syndrome” is well organized, constructed and written. The article deals with prognostic biomarkers of COVID-19 that could be used for risk evaluation and taking decision for adequate treatment of patients, diagnosed withSARS-CoV-2. This topic is very serious and the study is particularly important and relevant due to the need to apply personalized treatment algorithms according to the health status and characteristics of the individual patients. The authors found that resistin and IL-15 could be used as early predictors for patients at high risk required invasive ventilation.

I have some remarks.

  1. Please, check again the values in Table 2 for the cytokines IL-1b (page 7, last row) and IL-13 (page 8, row 5). Taking in account the values 4.91/4.94 and 14.93/12.86 it is impossible to have p-value<0.001.
  2. The same is valid for the Lymphocytes (1.02/1.31 – p=0.009, page 9, row 8) and IFN-beta (1386.98/299.07 – p=0.184, should be more significant) in Table 3.
  3. Page 13, line 348: “…as shown Figure 3” should be “as shown in Figure 3”.

Reviewer 2 Report

The manuscript entitled "Resistin and IL-15 as predictors of mechanical ventilation in COVID-19 pneumonia irrespective of the presence of obesity and metabolic syndrome" aims to assess the utility of resistin, IL-15, and other cytokines, as potential predictors of invasive or non-invasive ventilatory support in COVID-19 patients. Moreover, the authors studied the possible impact of obesity and metabolic syndrome on predicting mechanical ventilation needs in COVID-19. In my opinion, the manuscript raises major and minor concerns that authors should address in a point-by-point fashion before the article can be accepted for publication.

Points of criticism.

  1. It would be nice to define IL-15 and resistin in the introduction section.
  2. In my opinion, categorizing patients needing invasive or non-invasive mechanical ventilatory support as those with moderate COVID-19 is wrong. A lot of literature demonstrates that patients in need of invasive ventilatory support have an increased mortality risk.
  3. It is unclear when the authors took blood samples to run the multiplex assay. If they took blood samples 12 hours after hospital admission, I could expect that all enrolled patients had received different drug cocktails to treat COVID-19. So, how can we be sure that cytokine levels did not result from treating patients with tocilizumab, dexamethasone, or any other drug?
  4. In my opinion, the sample size appears not enough when subgrouping patients by comorbidities such as metabolic syndrome. Would you please provide details regarding sample size calculation? If the sample size is not enough, the authors should discuss it.
  5. It would be nice to know whether 34.9% of patients with remaining COVID-19 symptoms had changes in the cytokine profile compared to patients without symptoms.
  6. Please provide r and P values in the 3.6 result subsection.
  7. The authors used the CRT method to provide predictive values. However, what is relevant to clinical practice are sensibility and specificity values by the Youden index. Furthermore, why did the authors not use any more categorization by disease severity (mild, moderate, and severe)?
  8. Please indicate how did you calculate the risk of invasive mechanical ventilatory support? Relative risk? Odd-ratio?
  9. In ROC curves, please provide the n per group and CI values.
  10. Why did the authors limit their findings to the need for invasive or non-invasive ventilatory support? There are other relevant outcomes in the clinical practice, such as mortality or intensive care unit admission. For instance, what is the AUC of resistin or IL-15 for mortality risk?
  11. Please provide some insights explaining why all cytokine levels increased (except for IL-1 beta) in the discussion section.
  12. One of the manuscript's main goals is to study baseline cytokine levels in COVID-19 patients with obesity or metabolic syndrome. However, the authors did not show a subanalysis concerning the effect of obesity or metabolic syndrome on the risk of needing invasive or non-invasive ventilatory support.

Round 2

Reviewer 2 Report

I read the authors’ replies to the previous review report with great interest. Although the authors replied to the last comments, they still need to address additional concerns.

  1. I still consider that definitions of IL-15 and resistin should be first in the introduction section. I understand the author’s opinion regarding not commenting on the results in the introduction section. However, IL-15 and resistin are even part of the article’s title, and a proper definition would have improved the understanding of the reason behind the study.
  2. Patients enrolled in the study have already received drugs that can change cytokine expression and levels. This notion is still the weakest article´s component. How can we be sure that IL-15 or IL-10 levels result from the severity of pneumonia or are a mere manifestation of the use of different drug schemes on each patient?
  3. Do you consider a sensitivity value of 88.9% and a specificity value of 48.8% as a relevant value for making clinical decisions?
  4. Thank you for providing CI values from ROC curves. However, the wide-ranging CI values indicate that the sample size is small (i.e., IL-8+IL-15, CI 0.60-0.82).
